# Pathophysiological Behaviour of the Climber’s Foot versus the General Population: A Prospective Observational Study

**DOI:** 10.3390/healthcare10050868

**Published:** 2022-05-08

**Authors:** Paula Cobos-Moreno, Álvaro Astasio-Picado, Beatriz Gómez-Martín

**Affiliations:** 1Nursing Department, University of Extremadura, 10600 Plasencia, Spain; pcmoreno@unex.es (P.C.-M.); bgm@unex.es (B.G.-M.); 2Nursing, Physiotherapy and Occupational Therapy Department, Faculty of Health Sciences, University of Castilla-La Mancha, Real Fábrica de Sedas, s/n, 45600 Talavera de la Reina, Spain

**Keywords:** foot, plantar, pressures, chronic, injuries, sport climbing, first radius

## Abstract

Sport climbing is becoming increasingly popular, with people of all types and ages practising it. The feet suffer a lot of pressure with the sport climbing gesture, which in the long run can produce alterations in the first metatarsophalangeal joint or in the first radius of the foot. **Objective**: To observe and quantify the behaviour of the foot in climbing subjects compared to a group of non-climbing subjects, comparing the pressures, first metatarsophalangeal joint and first radius of the foot. **Method**: This is a non-experimental and observational, cross-sectional, descriptive and prospective research. The study sample consisted of 105 subjects (42 males and 63 females). The control group consisted of 52 subjects and the climbing group consisted of 53 subjects. Different exploratory tests were carried out on all the subjects, such as: mobility of the metatarsophalangeal joint and first radius of the foot and the study of plantar pressures in different areas of the study. **Results**: No significant difference was found between left and right foot measurements (*p* > 0.05). The pressures of the same foot are significant, both at static and dynamic stages for both groups. The maximum pressure in the climbing group was under the first metatarsal head, while in the control group it was under the second metatarsal head. There were significant differences in the mobility of the first metatarsal joint and the first radius between the two groups. **Conclusion**: It can be seen that the group of climbers has less plantar pressure than the control group. They also have altered mobility of the first radius and the first metatarsophalangeal joint.

## 1. Introduction

The origins of climbing have little to do with what we know today, since the first climbers can be considered the scientists who, during the 18th and 19th centuries, discovered many of the most important mountains in the world [1]. However, it was in the 1980s that sport climbing as we know it today was born [2].

Sport climbing is a booming sport, where the number of participants is constantly increasing [3]. Around 1985, the first rock climbing competitions took place. In 2021, climbing will be considered as an Olympic discipline for the first time at the Olympic Games in Tokyo [4].

Efficient climbing requires good footwork; both legs and feet are designed to support body weight, arms are not. Despite the above statement highlighting the greater importance of the feet over the arms, studies focus more on upper limb injuries [5]. Learning to optimally position the feet reduces stress on the forearms and positions the body to efficiently reach the next grip point. In addition, paying attention to the feet is important to avoid possible injury in the event of a slip.

However, it should be noted that sometimes the error lies not in the technique but in the quality of the climbing foot, thus providing poor grip and complicating effective foothold [6]. 

Most rock climbing and alpine climbing used to be performed in mountaineering boots, which were made of leather and eliminated any sensitivity, as well as putting a lot of force on the toes to propel oneself [7]. Many of the injuries that appear on climbers’ feet are due to the result of using unnaturally shaped or undersized climbing shoes. The small toe box keeps the foot in an unstable, supinated position, and the thinness of the shoe causes sensitivity to the foot. Climbing shoes fit like a second skin, to obtain this fit, climbers accept pain during and after climbing [7,8]. In addition, the feet suffer a lot of pressure in the sport, mostly due to the use of climbing shoes. This excess compression can eventually lead to alterations of the first metatarsophalangeal joint or the first radius of the foot [9,10]. There are several scientifically validated tools available to clinically assess these changes such as the goniometer [11,12], the first radius gauge [12] or pressure platform [13].

There is little bibliographic reference that talks about foot injuries due to the practice of climbing. There are recent publications describing injuries related to the use of climbing shoes. Traumatic injuries are the most common [14]. Injuries can be classified as acute or chronic. The most common acute injuries are: contusion, calcaneal fracture, talus fracture, ankle fracture, ankle sprain with lateral ligament injury, etc. [7]. While the most common chronic injuries are: hyperkeratosis, helomas, claw toes, Achilles tendon insertion injuries, fasciopathies. In addition to, the existence of pain in the toenails, as well as subungual haematomas, dystrophic nails, nail onycholysis, hallux valgus, hallux rigidus, nail infections, blisters, etc. [7,15].

The mechanism of foot injury depends on the technique used; in heeling, the heel is positioned on the support and force is exerted on it to rise above it by flexing the hamstrings. It is during this gesture that an external rotation of the knee is exerted, exerting great tension on its posterior and lateral structures. It could be said that the force exerted on the aforementioned structures, added to the angulation to which the knee is subjected, is what causes this type of injury during the heeling gesture. While in the frog position, the knees will be fully flexed and with a lateral orientation. The most common injuries in this type of position are meniscus tears, on the other hand, falls can be related to both upper and lower body joint injuries. The referred to lower body joint injuries vary depending on the climbing modality. Thus, in sport climbing, injuries usually occur because the climber hits the wall after falling, doing so with the knees extended [7,8].

Baropodometry is a technique that allows us to analyse the gait using devices such as baropodometers, allowing us to know the pressures exerted on each of the points of the plantar surface, both statically and dynamically. It makes it possible to visualise in real time, during the development of the gait, both the load surface and the line formed from the centre of gravity or body thrust [16]. The electronic baropodometric system is not intended to replace clinical examination or other podiatric or radiological examinations, but is an effective complementary method that can help us to better understand foot disorders [17]. Several authors have confirmed through work carried out with electronic baropodometers that all the metatarsals bear loads and that these mainly fall on the central metatarsals [18,19]. Thanks to this diagnostic technique, some surgical techniques, such as those used for the treatment of hallux abductus valgus, have been modified by verifying the loss of support of the distal phalanx of the hallux [20], just as important aspects of the surgical results in calcaneal fractures have been verified [21].

In the field of research, its use is oriented towards quantitative analysis of the foot, whether static or dynamic [22,23]. Specifically in dynamics, it allows the study of gait from the kinetic point of view and can be complemented with kinematic methods for a more complete analysis [24]. It makes it possible to visualise in real time, during the development of the gait, both the load surface and the line formed from the centre of gravity or body thrust [25]. This technique makes it possible to analyse the subject in a non-invasive way and obtain rapid results with a high level of precision, reliability and repeatability [26,27,28], data that will be useful for studying how the climber’s foot behaves compared to the non-climbing group and for preventive podiatry against the injuries that appear in the foot when practising this sport [20].

Therefore, the main objective of this research is to observe how the climber’s foot behaves compared to a non-climbing group (control group), comparing both the plantar pressures of the first radius and the pressures of the first metatarsophalangeal joint, as previous scientific evidence has shown that these are the anatomical structures that are most affected by the practice of this sport.

## 2. Methods

### 2.1. Type of Design

This study follows the four main lines according to Hernández-Sampieri [29] and Argimon and Jiménez [30]. It is a non-experimental, observational, cross-sectional, descriptive and prospective study. Within the type of study design, there is a group of climbers (cases) and a group of non-climbers (controls), and we compared both groups with respect to the study factors. All members were randomly selected.

### 2.2. Sample Size Calculation

For the sample size calculation, we used the formula: n = (2S^2 (Zα + Zβ)^2)/d^2, a standard deviation (S) was applied based on a previous pilot study (= standard error IA; = standard error II; d = minimum difference you want to detect). From this it was concluded that at least 53 subjects were required for each group in order to be able to make a comparison meeting these requirements.

### 2.3. General Characteristics of the Sample

The study sample consisted of 105 people, a total of 210 feet (42 men and 63 women). The control group consisted of 52 people and the climbing group consisted of 53 people. The inclusion and exclusion criteria were applied to the participating population. In the group of climbers, the inclusion criteria were: practice sport climbing regularly (minimum two days per week and a duration of two years), be federated in the FEXME, be residents of the autonomous community of Extremadura and patients who have freely agreed to participate in the study and have signed the corresponding informed consent. Similarly, the inclusion criteria of the control group: being healthy, having a normal footprint, being residents of the autonomous community of Extremadura. For both groups, the exclusion criteria were having a health problem or active pain in the lower limb. The study was carried out between January 2021 and November 2021. It was carried out in different types, firstly, the non-climbing population was recruited in the facilities of the university podiatry clinic in Plasencia, and then the climbers were recruited in the Cerezwall climbing wall in Plasencia.

### 2.4. Equipment and Procedure

Anthropometric data: age, sex, height and weight were recorded at the beginning of the study and collected on the recording sheet. To avoid differences in pressure measurements, all participants performed the test barefoot. The study of plantar pressures was carried out with the Podoprint^®^ pressure platform (Namrol Group, Barcelona, Spain). This system consists of a portable platform, with dimensions of only 570 mm × 570 mm and 9 mm thick and weighing 3.8 kg, which allows the user to carry out complete static and dynamic studies thanks to its 1600 sensors.

This platform has high sensitivity sensors capable of capturing 200 images per second and transmitting them to the connected computer via the Wi-Fi network. The captured images are processed by the manufacturer’s own software for Windows^®^.

The platform was set up on a 20-metre long, flat, unobstructed surface, allowing subjects to walk at their own walking pace, at a constant pace and without space concerns that could impair data recording or distort the subject’s gait [22]. All participants were instructed to walk the walkway as naturally as possible to avoid biases in each patient’s own gait. Participants undertook a pilot test prior to the start of the 10-min data collection session [8,22]. To avoid sampling bias due to the use of footwear, all participants performed the test barefoot. At the beginning of each sampling session the equipment was calibrated according to the manufacturer’s guidelines. Capturing the plantar pressures starts when the patient walks naturally, recording eight steps at random in each session (four steps with the right foot and four steps with the left foot), a sufficient number to provide reliability for the plantar pressures [8,9,10,19], of the four steps that come out of the multi-step system, we will take the clearest and most accurate image for the analysis of the image and obtaining the data necessary for the study.

Three different records are obtained in order to calculate the arithmetic mean and minimise intra-explorer bias [23,24]. All records were collected by the same observer. After each measurement, a check was made to ensure that the recording had been completed correctly, thus avoiding possible errors or false supports. Once it was correct, the next recording was continued. The record was considered valid and reliable only if there were at least two complete footprints for each foot.

For the assessment of the first radius, a scientifically validated first radius mobility meter was used to quantify the maximum degree of plantar and dorsal flexion in millimetres of this joint [7]. For this purpose, the subject was in the supine decubitus position on the examination table with the ankle relaxed and the subtalar joint in neutral position. With one hand, the explorer held the long limb over the heads of the second to fifth metatarsals, and with the other hand he held the short limb over the head of the first metatarsal. In this position, we moved the head of the first metatarsal to its maximum dorsal flexion and then to its maximum plantar flexion, recording the millimetres it moved in both directions. The measurement was taken three times for each subject in order to establish an arithmetic mean when studying this variable.

For the assessment of the metatarsophalangeal joint, a two-arm goniometer was used as a scientifically validated instrument [25]. The exploratory technique was performed according to the scientific literature using the technique indicated for this purpose [26]. As in previous exploratory techniques, three records were collected in order to be able to calculate the mean value when studying the variable.

The technique was carried out by two different explorers, one with 5 years of experience, while the second has more than 20 years of experience. Both scouts carried out several tests prior to the final data collection in order to reduce inter- and intra-scouting bias.

### 2.5. Ethical Considerations of the Study

All subjects signed the informed consent form and voluntarily agreed to participate in the study. This study was approved by the Bioethics Committee of the University of Extremadura, Cáceres (Spain) and was planned and conducted in accordance with the ethical principles of the Declaration of Helsinki. It was approved by the committee on 3 March 2021 with approval number 15/2021.

### 2.6. Data Processing and Statistical Analysis

All data were entered from paper surveys into an Excel worksheet using Microsoft Excel 2010. SPSS software version 21.0 for iOS^®^ was used for statistical analysis.

Means and standard deviations of variables were calculated for descriptive analysis. The non-parametric tests used were Wilcoxon signed-rank test, Friedman test and Mann–Whitney U-test. To assess inter-rater reliability, the intraclass correlation coefficient (ICC) was calculated. It was considered significant for the statistical analyses in this study and for the reliability of the Podoprint^®^ platform, *r* > 0.8^31^.

## 3. Results

The sample for this study consisted first of 120 randomly selected individuals. Two groups were established: the control group and the climbers’ group. Data were collected by two different scouts. Fifteen subjects were excluded as they did not meet all the inclusion criteria, so that the final sample comprising the study was 105 (42 men and 63 women). The mean age of the sample is 25.53 ± 9.528 and body mass index 22.46 ± 3.51 (Table 1).

### 3.1. Comparison of Plantar Pressures between the Control Group and the Group of Climbers

Firstly, it was found that the study data do not follow a normal pattern (*p*-values less than 0.05, Kolmogorov-Smirnov test), therefore, non-parametric statistical tests were chosen.

The mean values of peak pressures of the three study regions (first metatarsal head, second metatarsal head and ball of the first toe) can be seen in Table 2, values given in both static and dynamic. No significant difference was found between the measurements of the left foot versus the right foot (*p*-value > 0.05, Wilcoxon signed-rank test), which means that both feet behave the same.

In any case, the differences observed in Table 2, between the pressures of the same foot are significant (*p*-values less than one per thousand, Friedman Test), both in the static and dynamic of both groups.

In the climbing group, the maximum pressures located in the first metatarsal head were 2201.34 ± 96.31 g/cm^2^ in static and 2007.57 ± 93.50 g/cm^2^ in dynamic. In the control group, the maximum pressures located in the second metatarsal head in static and dynamic were 2445.75 ± 73.28 g/cm^2^ and 2326.48 ± 72.12 g/cm^2^, respectively. There was a significant difference between static and dynamic within the same group (*p*-value < 0.05, Wilcoxon signed-rank test). Pressures were higher in both the climber and control groups in the static than in the dynamic. (Table 3).

When comparing the climbers’ group with the control group, it can be seen that there is a significant difference between the two groups (*p*-value is less than 0.05, Mann–Whitney U-test), both in static and dynamic. In the climbing group the maximum pressure is in the first metatarsal head (53% of the climbers had the maximum pressure in the first metatarsal head, followed by 26% in the second metatarsal head), while in the control group the maximum pressure is in the second metatarsal head (77% of them had the pressure in the second metatarsal head, followed by 10% in the first ball of the foot) Table 4.

The lowest value obtained for the intraclass correlation coefficient for intra-observer reliability was 0.911, suggesting that the reproducibility of the measurement procedure was good.

### 3.2. Comparison of the Mobility of the First Radius between the Control Group and the Group of Climbers

The dorsal, plantar and total movement of the first radius in the sagittal plane has shown that the values between the left and right foot behave in the same way (*p*-value greater than 0.05, Wilcoxon sign rank test). Where differences are found is between groups, observing that the climbing group has a greater dorsal flexion than the control group, but the total range of movement in the climbing group is less than in the control group (*p*-value is less than 0.05, Mann–Whitney U-test) (Table 5).

### 3.3. Comparison of the Mobility of the First Metatarsophalangeal Joint between the Control Group and the Group of Climbers

The mobility of the first metatarsophalangeal joint was found to behave equally between the left and right foot (*p*-value greater than 0.05, Wilcoxon sign-rank test). Where differences are found is between groups, observing that the climbing group has less dorsal flexion and plantar flexion than the control group (*p*-value is less than 0.05, Mann–Whitney U-test) (Table 6).

## 4. Discussion

Firstly, we will proceed to assess the plantar pressures of the control group (non-climbers) compared to the case group (climbers), followed by the behaviour of the mobility of the first radius, and finally the mobility of the first metatarsophalangeal joint.

In accordance with the first objective, there are some recent studies in the scientific literature that address research topics related to the foot using the Podoprint platform [31,32,33,34,35]. However, none have studied plantar pressures in the climber. This study identifies the normal values of plantar pressures of the foot in three study areas in a healthy population versus a climbing population. This contribution to the scientific community can be considered a strength of the study, as the values identified can serve as a reference for establishing pathological values through future lines of research.

Those authors who have lines of research on plantar pressures studies state that measurements can be affected by physiological changes in muscle activity, posture and gait speed [23,35,36]. Therefore, it is not sufficient to use a single test to obtain dynamic foot parameters from a sample. By averaging several tests, the variability of gait patterns is reduced [37]. Other authors have suggested that three recordings are sufficient to obtain a consistent result [33]. In this study, the protocol was followed and as outlined in previous scientific publications in studies with similar interventions. Each of them, in turn, recorded four trials of each foot, (this being possible thanks to the option provided by the manufacturer of the Podoprint system: “multi-step”).

The most reliable plantar pressure parameter is the maximum pressure [24,38]. In the present study, in the non-climbing group (control group) the maximum pressure (Px) is found in the forefoot region (exactly under the head of the second metatarsal), a fact that coincides with the studies carried out by Xu C, who used the footscan [24] or the study by Maetzler [38], which used the Emed platform, both authors also looked for normal pressure parameters in healthy feet, the same objective as in this study. On the other hand, in the case group (climber group) the Px is located below the first metatarsal head. The Px of the second metatarsal (control group) is 2445.57 g/cm^2^ (239.68 Kpa) and for the climber group 2201.34 g/cm^2^ (215.73 Kpa), these values are lower than those found by the previously mentioned authors 367 Kpa, and 443 Kpa, respectively [24,38]. This may be due to the difference of sensors on the pressure platforms. These findings allow us to expose what is the normal distribution of pressures in the different areas of the foot for healthy subjects [19,28]. It is important to note that there are no studies that refer to plantar pressures in the climber’s foot, observing a maximum pressure lower than normal, which is indicative of possible injuries [19,31,32,33], since the higher the pressure, the greater the overload and the greater the possibility of fracture or alteration of the metatarsal heads. Other authors such as Martínez Nova [28], Casto MPD [32] or [31] have validated pressure platforms with the aim of making them useful for study within the branch of podiatry, which is also achieved with this pressure platform.

Another objective of the study was to compare and quantify the mobility of the first radius between the feet of healthy patients (non-climbers) and the feet of climbers. The results obtained reveal that climbers had a lower range of motion (ROM) than non-climbers, but that dorsal flexion was greater. Information that we can corroborate with subsequent studies [12,39,40]. In the 2018 study [15], they showed that the movement of the first radius was 12.66 mm, with this movement being 3 mm greater in both the control group and the climbers in our study (9.31, 9.80, respectively), which may be due to the fact that in that study mobility was studied by means of X-rays, a fact that also happens in the 2021 study [39]. In the 2020 study [12] only 1 mm difference was found, which may be due to the fact that in both studies the same device was used to measure mobility. Dorsiflexion and plantarflexion were also significant, with less dorsiflexion observed in non-climbing patients than in climbers. Previous investigators also found a difference between normal patients versus patients with some impairment of the first radius [12,41,42,43].

Finally, when comparing the mobility of the first metatarsophalangeal joint, the data reveal that climbers have limited dorsal flexion of the first metatarsophalangeal joint, as the value is below the normal mean [44,45,46], which is necessary for good propulsion. The data from the climbers cannot be compared with other authors, as there are no previous studies with athletes in this discipline.

As for the limitations of the study, although the results obtained are conclusive in terms of the objectives of the study, larger samples could yield more conclusive results. The heterogeneity among the participants means that the results found should be taken with caution. This, in itself, justifies the implementation of future research.

On the other hand, the strong point of the study is the novelty of this study for the scientific population, as there are no studies that focus on comparing how a climber’s foot is compared to a normal foot. Knowing how a climber’s foot behaves cannot help to avoid possible pathologies or chronic alterations that appear as a consequence of practising this sport. These data can serve as a reference to establish common chronic injuries in the climber’s foot through future lines of research, that allow us to focus on the prevention of injuries to the climber’s foot, helping them to avoid them and improving their sports performance with foot control, which allows us to focus on the prevention of injuries to the climber’s foot, helping them to avoid them and improving their sports performance with foot control.

## 5. Conclusions

The climbers’ group had lower maximum pressure values than the control group. In the same way, it was shown that in the climbers’ group the maximum pressure was located under the first metatarsal head compared to the normal population that located it in the second metatarsal head. On the other hand, the first radius presented a decreased ROM in the climbing group compared to the control group, where an increase in dorsal flexion was observed. Additionally, finally, the first metatarsophalangeal joint was limited, as the values were well below the normal value, which is necessary for good propulsion of the foot.

These findings allow us to know how the climber’s foot behaves and seek preventive podiatry to improve both the practice of climbing and the life of the subject studied.

## Figures and Tables

**Table 1 healthcare-10-00868-t001:** Descriptive data of the study participants showing demographic and anthropometric characteristics according to sex.

	**Climbing Group**					
	MAN *n* = 32		WOMAN *n* = 21		Total, *n* = 53	
	MEAN ± SD (95% CI)	VALOR P	MEAN ± SD (I95% CI)	VALOR P	MEAN ± SD (95% CI)	VALOR P
AGE	29.22 ± 2.21 (24.69–33.74)	0.200	25.00 ± 2.87 (19.01–30.99)	0.078	27.55 ± 1.76 (24.01–31.09)	0.051
BMI	22.49 ± 0.41 (21.36–23.62)	0.200	19.76 ± 0.74 (18.20–21.31)	0.200	21.41 ± 0.32(20.45–22.37)	0.200
**Control Group**						
	MAN *n* = 10		WOMAN *n* = 42		Total. *n* = 52	
	MEAN ± SD (95% CI)	VALOR P	MEAN ± SD (I95% CI)	VALOR P	MEAN ± SD (95% CI)	VALOR P
AGE	23.50 ± 0.93 (21.39–25.61)	0.014	23.48 ± 0.49 (22.55–24.40)	0.010	23.48 ± 0.40 (22.66–24.30)	0.001
BMI	24.00 ± 0.31 (23.30–24.70)	0.104	23.42 ± 0.54 (22.32–24.53)	0.200	23.53 ± 0.45(22.64–24.43)	0.200

SD = Standard deviation; BMI = Body mass index; 95% CI = 95% confidence interval European countries.

**Table 2 healthcare-10-00868-t002:** Mean maximum pressure in the study population by study groups and different foot.

		**CLIMBING GROUP**						
		**LEFT FOOT**	**RIGHT FOOT**	**P-VALOR**	**DYNAMIC**	**LEFT FOOT**	**RIGHT FOOT**	**P-VALOR**
		**MEAN ± SD (95% CI)**	**MEAN ± SD (I95% CI)**			**MEAN ± SD (95% CI)**	**MEAN ± SD (I95% CI)**	
**STATIC**	**PX 1MT** g/cm^2^	2201.34 ± 96.31 (2006.06–2394.62)	2211.58 ± 107.91 (1995.04–2428.13)	0.873		2007.57 ± 93.50 (1819.93–2195.20)	2105.06 ± 98.42 (1907.57–2302.55)	0.176
	**PX 2MT** g/cm^2^	2020.75 ± 98.15 (1823.80–2217.71)	2010.98 ± 96.14 (1817.43–2204.35)	0.825		1887.08 ± 84.22 (1717.95–2056.20)	1853.23 ± 87.86 (1676.92–2029.53)	0.732
	**PXP** g/cm^2^	1730.02 ± 101.34 (1526.66–1933.38)	1812.85 ± 102.81 (1606.53–2019.17)	0.197		1505.95 ± 99.58 (1306.01–1705.68)	1572.13 ± 101.308 (1368.84–1775.42)	0.502
**CONTROL GROUP**								
**STATIC**	**PX 1MT** g/cm^2^	1550.10 ± 73.44 (1402.66–1697.53)	1538 ± 68.63 (1401.18–1676.74)	0.690	**DYNAMIC**	1435.98 ± 64.87 (1305.74–1566.22)	1425.94 ± 51.89 (1321.77–1530.12)	0.970
	**PX 2MT** g/cm^2^	2445.75 ± 73.28 (2298.63–2592.87)	2479.62 ± 68.12 (2343.62–2616.39)	0.817		2326.48 ± 72.12 (2181.68–2471.28)	23778.21 ± 60.47 (2256.79–2499.63)	0.974
	**PXP** g/cm^2^	1287.75 ± 86.67 (1113.78–1461.72)	1277.81 ± 93.79 (1089.52–1466.10)	0.731		1101.12 ± 80.94 (938.61–1263.62)	1127.44 ± 85.86 (955.06–1299.82)	0.873

PX1MT = Maximum pressure first metatarsal; PX2MT = Maximum pressure second metatarsal; PXP = Maximum pressure first pulp; SD = Standard deviation;.95% CI = 95% confidence interval European countries.

**Table 3 healthcare-10-00868-t003:** Maximum static and dynamic pressure.

		STATIC	DYNAMIC			STATIC	DYNAMIC	
**CLIMBING** **GROUP**		MEAN ± SD (95% CI)	MEAN ± SD (I95% CI)	P-VALOR		MEAN ± SD (95% CI)	MEAN ± SD (I95% CI)	P-VALOR
	**PX1MT** g/cm^2^	2206.46 ± 706.13 (1012.50–3759.50)	2056.31 ± 655.48 (1048–3883.5)	0.000	**CONTROL GROUP**	1544.53 ± 487.02 (777–3204.5)	1430.96 ± 393.04 (515.5–2832)	0.000
	**PX2MT** g/cm^2^	2015.82 ± 696.96 (863–3839)	1870.15 ± 612.07 (926–3715)	0.00		2462.68 ± 482.03 (1521–3497.5)	2352.35 ± 440.75 (1512–3411.5)	0.00
	**PXP** g/cm^2^	1771.43 ± 708.83 (649–3444)	1538.99 ± 700.26 (589–3395)	0.000		1282.77 ± 629.33 (400–2877.5)	1114.27 ± 580.67 (362–2840.5)	0.000

PX1MT = Maximum pressure first metatarsal; PX2MT = Maximum pressure second metatarsal; PXP = Maximum pressure first pulp; SD = Standard deviation; 95% CI = 95% confidence interval European countries.

**Table 4 healthcare-10-00868-t004:** Percentage of occurrence of plantar pressures.

		CLIMBING GROUP		CONTROL GROUP	
		STATIC	DYNAMIC	STATIC	DYNAMIC
PERCENTAJE	PX1MT	53	61	8	4
	PX2MT	27	20	77	85
	PXP	20	19	10	8
	PX4MT	0	0	5	3

PX1MT = Maximum pressure first metatarsal; PX2MT = Maximum pressure second metatarsal; PXP = Maximum pressure first pulp; PX4MT = Maximum pressure quarter metatarsal.

**Table 5 healthcare-10-00868-t005:** Movement of the first radio of all participants.

	CLIMBING GROUP			CONTROL GROUP			P VALOR *
	LEFT FOOT	RIGHT FOOT		LEFT FOOT	RIGHT FOOT		
	MEAN ± SD (I95% CI)	MEAN ± SD (I95% CI)	P-VALOR	MEAN ± SD (I95% CI)	MEAN ± SD (I95% CI)	P VALOR	
MFD	5.56 ± 1.07 (3–8)	5.43 ± 1.07 (3–8)	0.791	4.78 ± 1.51 (2–12)	4.55 ± 1.63 (2–12)	0.324	0.007 *
MFP	3.75 ± 1.37 (2–10)	3.35 ± 1.21 (2–9)	0.166	4.26 ± 1.31 (2–9)	4.15 ± 1.21 (1–6)	0.102	0.000 *
MT	9.31 ± 2.12 (4–17)	8.88 ± 2.22 (3–15)	0.100	9.80 ± 1.54 (6–16)	9.94 ± 1.52 (7–13)	0.709	0.001 *

MFD = Movement in dorsiflexion; MFP = Movement in plantarflexion; MT = Total movement; SD = Standard deviation; 95% CI = 95% confidence interval European countries; * *p* value between scaling group and control group.

**Table 6 healthcare-10-00868-t006:** Movement of the first metatarsophalangeal joint of all participants.

	CLIMBING GROUP			CONTROL GROUP			P VALOR *
	LEFT FOOT	RIGHT FOOT		LEFT FOOT	RIGHT FOOT		
	MEAN ± SD (I95% CI)	MEAN ± SD (I95% CI)	P-VALOR	MEAN ± SD (I95% CI)	MEAN ± SD (I95% CI)	P VALOR	
FD	53.85 ± 12.42 (10–76)	54.14 ± 10.81 (25–80)	0.902	70.40 ± 11.65 (37–80)	70.07 ± 11.43 (40–89)	0.919	0.000 *
FP	29.07 ± 8.14 (10–50)	29.15 ± 8.04 (10–50)	0.867	38.07 ± 8.24 (19–59)	37.76 ± 8.10 (19–59)	0.915	0.000 *

FD = Dorsiflexion; FP = Plantarflexion; SD = Standard deviation; 95% CI = 95% confidence interval European countries; * *p* value between scaling group and control group.

## Data Availability

Not applicable.

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
