# Peer review of "Pathophysiological Behaviour of the Climber’s Foot versus the General Population: A Prospective Observational Study"

_healthcare, 2022, doi:10.3390/healthcare10050868_

Round 1
Reviewer 1 Report
The authors of the article set the following objective: observe and quantify the behaviour of the foot in climbing subjects compared to a group of non-climbing subjects, comparing the pressures, first metatarsophalangeal joint and first radius of the foot.
My comments and suggestions for changes to the article are the following:
1º Indicate the type of study performed in your title and abstract.
2º The number of keywords defining your study is scarce, introduce a minimum of 5 keywords and if possible make them MESH terms.
3º You could attach the STROBE checklist for verification, since in the methodology you indicate that it is an observational study and I have detected that some items are not taken into consideration. It would be advisable for you to follow the indications of STROBE.
4º They must provide prevalence data with current and justified literature.
5º Explain and expand information in your introduction about the mechanism of injury that climbing can cause to the climber's foot with current and relevant bibliographic references.
6º Indicate how the sample size has been calculated.
7º Describe within the methodology the relevant setting, locations and dates, including recruitment, exposure, monitoring and data collection periods. data collection.
8º Observational studies are characterized by a detailed description of the participants, which is neither present in the selection criteria, which are not defined (age range, sex, height, weight socioeconomic level, level of physical activity, etc.), nor in the form of a descriptive table in the results (Table of characteristics of the participants).
9º In the ethical considerations of the study should provide the approval number of the ethics committee not the date of registration of when it was performed.
10º Define in the methodology section in isolation (headings and subheadings may be used) the variables analyzed; data sources or measures used and measures taken to address potential sources of bias.
11º Delete the following expression: "Then, following the same structure established in the presentation of the results, we proceed to the discussion of the results". You can use a prospective to the objective of the study to start the discussion.
12º Discussions should cover the key findings of the study: discuss any previous research related to the topic to place the novelty of the discovery in the appropriate context, discuss possible shortcomings and limitations in its interpretations, discuss its integration into the current understanding of the problem and how. This advances current views, speculates on the future direction of research, and freely postulates theories that could be tested in the future, completed, and reformulated.
They should be more explanatory and descriptive in their discussion, the information at some points in their discussion is incomplete, see some examples:
"Although the finding that the peak pressure is below the second metatarsal head is consistent with what other authors have shown in their studies using previously validated pressure platforms other than the Podoprint system (8,10)". What do these studies indicate regarding their results.
"Regardless of this finding, most authors agree that the forefoot has the highest values of plantar pressures recorded (9,10,12,32)." What do these four studies indicate in reference to their results.
"Which is indicative of possible injury (19,31,33)." because it is indicative of possible injury.
They should reformulate their discussion and expand the content with current literature.
13º The limitations of the study should be rephrased as this research has some bias (comparing subjects with excessive exposure to a traumatic action with healthy subjects with no exposure).
They indicate that their strong point is: “On the other hand, the strength of the study is the novelty of this study for the scientific population, as there are no studies that focus on comparing how a climber's foot is compared to a normal foot, helping to avoid possible pathologies or chronic alterations that appear as a consequence of practising this sport.” but if we find bibliographical references in this regard which should be taken into consideration for the discussion and introduction of this manuscript:
Identeg F, Orava E, Sansone M, Karlsson J, Hedelin H. Patterns of traumatic outdoor rock-climbing injuries in Sweden between 2008 and 2019. J Exp Orthop. 2021 Oct 9;8(1):89. doi: 10.1186/s40634-021-00407-1. PMID: 34628554; PMCID: PMC8502181.
Buda R, Di Caprio F, Bedetti L, Mosca M, Giannini S. Foot overuse diseases in rock climbing: an epidemiologic study. J Am Podiatr Med Assoc. 2013 Mar-Apr;103(2):113-20. doi: 10.7547/1030113. PMID: 23536501.
Hawrylak A, Chromik K, Ratajczak B, Barczyk-Pawelec K, Demczuk-Włodarczyk E. Spinal range of motion and plantar pressure in sport climbers. Acta Bioeng Biomech. 2017;19(2):169-173. PMID: 28869622.
14º Conclusions should have their own section.
15º You should introduce current literature in a justified way and allow discussion of the results, you present a large number of citations that exceed 10 years old both in their introduction and in their discussion, examples references (2, 6, 10, 11, 12, 15, 17, 18, 21, 23, 28, 29, 31, 38) and others exceed 20 years old (3, 8, 35, 36).
16º The bibliographic references 16 and 37 do not present sufficient methodological quality for a research of these characteristics and for a journal of this impact.
17º They should use different bibliographic references in their introduction and discussion section, it is not correct to use the same bibliographic citations (they should analyze more deeply the subject of study).
18º Include in the conclusions a prospective for future researchers in reference to the limitations found or possible deficiencies of the research proposed.
Author Response
Dear reviewer,
We appreciate your assessment of the manuscript and greatly appreciate major and minor suggestions to enrich it.
As for the main points:
1. Indicate the type of study conducted in your title and abstract: Pathophysiological Behavior of the Climber's Foot vs. The General Population. ObservationalStudio.
Method: It is a non-experimental and observational, cross-sectional, descriptive and prospective research.
2. The number of keywords that define your study is scarce, enter a minimum of 5 keywords and, if possible, MESH terms: Foot, Plantar, Pressures, Chronic, Injuries, Sport Climbing, First Radius
3. They must provide prevalence data with updated and justified bibliography: At present, we have only found this study 15. Cobos-Moreno P, Astasio-Picado Á, Gómez-Martín B. Epidemiological Study of Foot Injuries in the Practice of Sport Climbing . 2022; Available from: https://doi.org/10.3390/ijerph19074302, which deals with chronic foot pathologies, hence the importance of this study.
4. Explain and expand the information in your introduction on the mechanism of injury that climbing can produce in the climber's foot with current and relevant bibliographical references: The mechanism of injury to the foot depends on the technique used, in heeling the heel is positioned on the support and force is exerted on it to rise above it by flexing the hamstrings. It is during this gesture that an external rotation of the knee is exerted, exerting great tension on its posterior and lateral structures. It could be said that the force exerted on the aforementioned structures, added to the angulation to which the knee is subjected, is what causes this type of injury during the heeling gesture. While in the frog position the knees will be fully flexed and with a lateral orientation. The most common injuries in this type of position are meniscus tears, on the other hand, falls can be related to both upper and lower body joint injuries. Referred lower body joint injuries vary depending on the climbing modality. Thus, in sport climbing, injuries usually occur because the climber hits the wall after falling, doing so with the knees extended (7-8).
5. Indicate how the sample size was calculated: To calculate the sample size, the formula was used: n= (2S^2 (Zα+Zβ)^2)/d^2, a standard deviation (S ) based on a pilot study prior to this one. (α= standard error IA; β= standard error II; d= minimum difference you want to detect). With this, it was concluded that at least 53 subjects were required as a minimum for each group to be able to make a comparison fulfilling these requirements.
6. Describe within the methodology the relevant setting, locations, and dates, including periods of recruitment, exposure, monitoring, and data collection. Data collection: The study was carried out between January 2021 and November 2021. It was carried out in different types, firstly, the non-climbing population was recruited, in the facilities of the Plasencia university podiatric clinic, and then they recruited Lao climbers, at the Cerezwall climbing wall in Plasencia
7. Observational studies are characterized by a detailed description of the participants, which is not present in the selection criteria, which are not defined (age range, sex, height, weight, socioeconomic level, level of physical activity, etc.). ), nor in the form of a descriptive table in the results (Table of characteristics of the participants): The study is of 105 (42 men and 63 women). The mean age of the sample is 25.53 ± 9.528 and a body mass index of 22.46 ± 3.51. Weight and height have been unified in the BMI, socioeconomic level is not of interest to us for this study.
8. In the ethical considerations of the study, the approval number of the ethics committee must be recorded, not the date of registration of when it was carried out: It was approved by the committee on March 3, 2021, whose approval number is 15/2021.
9. Define in the methodology section in isolation (titles and subtitles can be used) the variables analyzed; data sources or measures used and measures taken to address possible sources of bias: this section has been rewritten following your instructions by adding the subtitles:
design type
sample size calculation
general characteristics of the sample
equipment and procedure
ethical considerations of the study
data processing and statistical analysis
10. Suppress the following expression: “Then, following the same structure established in the presentation of the results, the results are discussed”. You can use a prospective study objective to start the discussion: We've deleted it.
11. Discussions should cover the key findings of the study: discuss any previous research related to the topic to place the novelty of the discovery in the proper context, discuss possible shortcomings and limitations in their interpretations, discuss their integration into the current understanding of the problem, and as. This advances current views, speculates on the future direction of research, and freely posits theories that could be tested in the future, supplemented, and reformulated: The discussion has been redone with your considerations.
12. The limitations of the study should be reformulated since this research has some bias (comparing subjects with excessive exposure to a traumatic action with healthy subjects without exposure): Regarding the limitations of the study, it has been found that: larger samples could yield more conclusive results. The heterogeneity between the participants means that the results found should be taken with caution. This in itself justifies the implementation of future research.
13. They indicate that their strong point is: “On the other hand, the strength of the study is the novelty of this study for the scientific population, since there are no studies that focus on comparing how the foot of a climber compares with a foot normal, helping to avoid possible pathologies or chronic alterations that appear as a consequence of the practice of this sport.” but we do find bibliographical references in this regard that should be taken into account for the discussion and introduction of this manuscript: At present there is no study that compares non-climbering populations with climbers, if there are studies that speak of climber injuries but do not compare them with a non-climbing group.
14. The conclusions should have their own section: Fact.
15. You must introduce current literature in a justified way and allow the discussion of the results, you present a large number of citations that exceed 10 years both in their introduction and in their discussion, reference examples (2, 6, 10, 11, 12, 15, 17, 18, 21, 23, 28, 29, 31, 38) and others are over 20 years old (3, 8, 35, 36): The discussion has been redone using the most current references that have been found discussing with the ones that were there before.
16. Bibliographical references 16 and 37 do not present sufficient methodological quality for an investigation of these characteristics and for a journal of this impact: done.
17. Include in the conclusions a prospective for future researchers in reference to the limitations found or possible deficiencies of the proposed research: these data can serve as a reference to establish common chronic injuries in the climber's foot through future lines of research, which allow us to focus on the prevention of injuries to the climber's foot, helping them to avoid them and improve their sports performance with foot control.
We especially appreciate your great review of the article. We have proceeded to review it completely and in detail. We hope it is of your consideration.
Very thankful.
Reviewer 2 Report
Dear authors:
It has been a pleasure to review your paper about “Pathophysiological Behaviour of the Climber’s Foot vs. The General Population” but I have observed a few errors that it’s necessary to change it before to accepted it. You can see below the recommendation
Title: Please can you include the type of study
Method: How did you calculate the sample size?
Results: Please review the table 1 (p-valor), I think that will be p-value and what mean DSDE
Discussion: Can you improve the limitations and include a text with the clinical implications of this results?
Author Response
Dear reviewer,
We appreciate your assessment of the manuscript and we highly value major and minor suggestions to enrich it.
Regarding the main points:
1) Title: Please can you include the type of study:
Pathophysiological Behavior of the Climber's Foot vs. The General Population. ObservationalStudio.
2) Method: How did you calculate the sample size?: To calculate the sample size, the formula was used: n= (2S^2 (Zα+Zβ)^2)/d^2, a standard deviation ( S) based on a pilot study prior to this one. (α= standard error IA; β= standard error II; d= minimum difference you want to detect). With this, it was concluded that at least 53 subjects were required as a minimum for each group to be able to make a comparison fulfilling these requirements.
3) Results: check table 1 (p-value), I think it will be p-value and what does DSDE mean: Grammar error
4) Discussion: Can the limitations be improved and include a text with the clinical implications of these results? These data can serve as a reference to establish common chronic injuries in the climber's foot through future lines of research, which will allow us to focus on the prevention of injuries to the climber's foot, helping them to avoid them and improve their sports performance with foot control.
We have proceeded to review it completely and in detail.
We hope it is of your consideration.
Very thankful.
Reviewer 3 Report
Thank you for the opportunity to review the manuscript. Some points need to be further clarified. Are they:
1. Inclusion and exclusion criteria need to be better defined.
2. Are the instruments used for analysis sufficiently accurate? Do they have safe characteristics to carry out the evaluations of the variables?
3. Results should be guided by differences between groups. And not intra-group. Likewise the conclusion.
4. The discussion does not present any aspect of clinical applicability. Not even limitations of the study.
5. In fact, what are the main findings. And how these can impact researchers and clinicians.
Author Response
Dear reviewer,
We appreciate your assessment of the manuscript and we highly value major and minor suggestions to enrich it.
Regarding the main points:
1) It is necessary to better define the inclusion and exclusion criteria: The inclusion and exclusion criteria were applied to the participating population. In the group of climbers, the inclusion criteria are: Practice sport climbing regularly (minimum two days per week and two years old), be federated in the FEXME, be residents of the autonomous community of Extremadura and patients who have freely agreed to participate. in the study and have signed the corresponding informed consent. Similarly, the inclusion criteria of the control group: being healthy, normal foot print, being residents of the autonomous community of Extremadura. For both groups, the exclusion criteria were having some health problem or active pain in the lower limb.
2) Are the instruments used for the analysis accurate enough? Do they have safe characteristics to carry out the evaluations of the variables?: The instruments used are totally precise and validated, in addition, three records were taken per subject studied, and these three studies were carried out by two different explorers, thus avoiding possible biases.
3) Results should be guided by differences between groups. And not intragroup. Likewise, the conclusion: When comparing the climbers group against the control group, it can be observed that, if there is a significant difference between both groups (p-value is less than 0.05, Mann-Whitney U Test), both in static and in dynamic. In the climbers group, the maximum pressure is in the first metatarsal head (53% of the climbers had the maximum pressure in the first metatarsal head, followed by 26% in the second metatarsal head), while in the control group the maximum pressure it is found in the second metatarsal head (77% of them presented pressure in the second metatarsal head, followed by 10% in the first pad) table 4.
4) The discussion does not present any aspect of clinical applicability. Not even the limitations of the study: The strong point of the study is the novelty of this study for the scientific population, since there are no studies that focus on comparing how the foot of a climber is compared to a normal foot. Knowing how the climber's foot behaves can help us avoid possible pathologies or chronic alterations that appear as a result of practicing this sport. These data can serve as a reference to establish common chronic injuries in the climber's foot through future lines of research, which will allow us to focus on the prevention of injuries in the climber's foot, helping them to avoid them and improving their sports performance with a foot control.
5) In fact, what are the main findings. And how these can affect researchers and doctors: The main findings are, as we say in the conclusions, how the climber's foot has limited mobility of the first joint and an increase in pressure in said joint, causing alterations in the foot such as they are HAV, limtus or painful overloads, whether in the climber's day-to-day or practicing climbing, hence the need to know how the scald foot behaves and how to avoid the appearance of these injuries.
We have proceeded to review it completely and in detail.
We hope it is of your consideration.
Very thankful.
Round 2
Reviewer 1 Report
Although some important errors have been solved, a revision by an English expert should be made, an example of which is the title of the manuscript:
1º It is an error to indicate "Observational Studio" it is more correct ":A prospective observational study".
2º It is not appropriate to use abbreviations in the title: replace "vs." with "versus". Proofreading by an English language expert is required.
3º Bibliographic references should be adapted to the journal format.
4º The reference 44 little scientific rigor, substitute.
Author Response
Dear reviewer,
We appreciate your reassessment of the manuscript and greatly appreciate major and minor suggestions to enrich it.
As for the main points:
1) Title, it is an error to indicate "Observational Study" it is more correct "Prospective observational study": modified.
2) The use of abbreviations in the title is not appropriate: replace "vs." with "against". English Expert Review Required: Modified.
3) Bibliographical references must be adapted to the format of the journal: revised and modified.
4) Reference 44 lacks scientific rigor, substitute: revised and replaced.
Similarly, we have proceeded to review it completely by an expert from the University's language department.
We hope it is of the consideration of him.
Very thankful.
Reviewer 3 Report
Congratulations to the authors for the clarification.
Author Response
Dear reviewer,
We appreciate your new assessment of the manuscript, as well as your consideration.
Very thankful.